# Training & Label-Free Domain Adaptation in 3D Object Detection for Autonomous Driving

## Abstract

Autonomous driving systems rely on 3D object detection for path planning and control, but it has been demonstrated that LiDAR-based 3D object detectors suffer a significant performance drop when tested on datasets from different geographic locations than their training data due to domain shift. Existing domain adaptation methods typically require access to original training data or substantial target domain data, making them impractical for resource-constrained deployments. In this work, we address this challenge by exploring how LiDAR-based 3D object detection models trained in a source domain can be reused in a data-scarce target domain without extensive fine-tuning. We propose lightweight and practical approaches for single-frame domain adaptation under three different resource-constraint settings: 1) a test-time approach that adjusts predictions using a calibration factor when only *a minimal amount of unlabeled target data* is available, 2) a lightweight adaptation technique using learnable scaling weights when *a small amount of labeled target data* are available, while preserving original model parameters, and 3) a weakly supervised finetuning method with a novel loss function for scenarios with *larger amounts of unlabeled target domain data*. Our results demonstrate that scalable, cross-regional deployment of 3D object detection systems is feasible under significant scarcity of data and training budget.

## 1 Introduction

Autonomous driving vehicles critically depend on the ability to accurately detect objects (cars, pedestrians, cyclists, etc.) and precisely localize them in 3D space. This object detection capability directly drives critical downstream tasks like path planning and collision avoidance. LiDAR-based 3D object detectors have emerged as the gold standard for this task. These detectors use point cloud data to predict accurate 3D bounding boxes around detected objects (Shi et al., 2019; Lang et al., 2019). The quality of these 3D object detection systems fundamentally determines the safety and reliability of autonomous driving systems. As autonomous vehicles' adoption and demand are increasing globally, the generalization ability of 3D detection models is becoming increasingly crucial for rapid deployment cycles without requiring fine-tuning these models for each new geographical region.

The extensive research, however, has demonstrated that LiDAR-based 3D object detectors exhibit poor cross-domain generalization – performance degradation reaching up to 40% when evaluated on target domains that differ from the training distribution. For instance, Wang et al. (2020b) showed a PointRCNN (Shi et al., 2019) trained on the KITTI dataset (Geiger et al., 2012; 2013) (German road dataset) experiences a performance drop of 36% when tested on the Waymo dataset (Sun et al., 2020) (US road dataset) as compared to when it was trained on the Waymo dataset itself. This degradation was primarily attributed to the size differences between vehicles across domains – American vehicles are typically larger in size than their German counterparts. Remarkably, correcting this vehicle size bias alone improved the car detection accuracy from 43% to 75%, representing a substantial 32% gain (Wang et al., 2020b).

Several domain adaptation techniques have been proposed to tackle such size dimension biases, including Statistical Normalization (SN) (Wang et al., 2020b), ST3D (Yang et al., 2021), and few-shot fine-tuning (Wang et al., 2020b). However, these techniques share a fundamental limitation – *they require access to either the original source training data or labeled target domain data, each then*

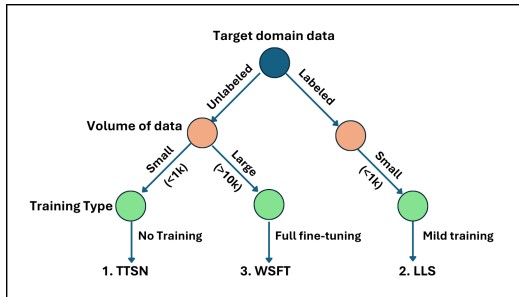

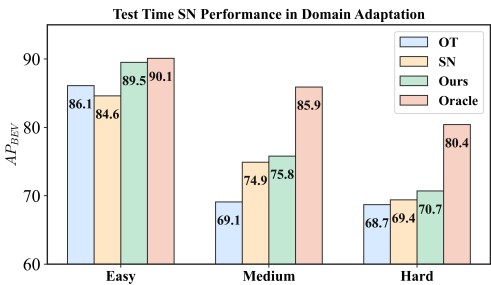

Figure 1: Choosing the right method based on data and training resource constraints.

Figure 2: TTSN beats SN and OT (Output Transformation) methods (KITTI → Waymo)

*followed by a mandatory model fine-tuning step.* This requirement presents challenges in two practically relevant scenarios. First, in data-scarce scenarios, where source domain data is proprietary or inaccessible, and collecting or annotating point cloud data in the target domain involves prohibitively high cost and time. Second, in deployment scenarios, where preserving the source model's representations is critical because fine-tuning risks catastrophic forgetting and thereby, degrading the model's ability to capture the original source domain distribution. These scenarios are increasingly common in practice, where open-source pretrained LiDAR-based 3D object detection models are readily available but their training data remains inaccessible, and the pretrained model serves as the repository of basic knowledge about object detection that can't be afforded to be forgotten.

In this work, we focus on the above resource constraint scenarios and propose a set of three simple yet effective methods (summarized below) for domain adaptation in 3D object detection. These methods provide targeted solutions across three distinct deployment scenarios (Figure 1). While designing these methods, we recognize that practical deployment constraints vary significantly based on available resources, data accessibility, and computational limitations.

1. **Test-Time Statistical Normalization (TTSN):** This method is useful when we have access to a limited amount ($\leq 1K$) of unlabeled data, as well as average car size statistics for the target domain. Further, this method is also useful when we can't fine-tune the source model due to either lack of computational resources or to prevent catastrophic forgetting of learned source domain representations, or both. This method is free from all of the following three resources: *source data, labels of target data*, and *model fine-tuning*. This method is based on the idea of using calibration factor at test time to better align predictions with target domain's ground truth. This method is shown to outperform SN (Wang et al., 2020b), which require fine-tuning the model on modified source data. Figure 2 highlights the performance of the TTSN method against popular baselines.

2. **Lightweight Linear Scaling (LLS):** This method is useful in cases where a small quantity ($\leq 1K$) of labeled target domain data is available but the fine-tuning still remains infeasible due to resource limitations or deployment requirements that prioritize model stability. This method employs a tiny regression network, called *domain-adapter*, to learn scaling factor for each dimension of the car size. This adapter network is designed to correct the size of predicted 3D bounding boxes accurately so as to fit them better in the target domain.

3. **Weakly-Supervised Fine-Tuning (WSFT):** This method is useful in environments where substantial amount ($\geq 10K$) of unlabeled target domain data and corresponding car size statistics are available, along with sufficient computational resources to permit full model fine-tuning and adaptation. This method is based on a novel loss function, called *Composite Statistical Alignment (CSA) loss*, which is designed to enable source data-free, weakly-supervised domain adaptation, facilitating the full fine-tuning of the source model without requiring the original source data.

Above three scenarios cover the full spectrum of real-world deployment challenges, where one need to balance three key factors: high cost of point-cloud annotation, limited training resources, and need to preserve source domain performance while adapting to new target domains. For each scenario,

the proposed method reduces the object mislocalization errors and improves the detection accuracy while being within the constraints.

## 2 RELATED WORKS

**LiDAR-based 3D Object Detection:** Existing 3D object detection work can be classified in three types based on the input data types: LiDAR-based (Shi et al., 2020; Lang et al., 2019; Shi et al., 2019; Yan et al., 2018), image-based (Chen et al., 2015; Xu & Chen, 2018; Li et al., 2024), and multi-modal fusion (Chen et al., 2017; Qi et al., 2018; Liang et al., 2022) methods, with LiDAR-based methods demonstrating superior performance for precise 3D localization. LiDAR-based detectors can further be divided roughly in three types – point-based methods operating directly on raw point clouds (Shi et al., 2019; Yang et al., 2020), voxel-based (Lang et al., 2019; Yan et al., 2018; Zhou & Tuzel, 2018) methods discretizing points into regular 3D grids, and hybrid point-voxel (Shi et al., 2020; Liu et al., 2019) methods combining both representations. Point-based detectors use PointNet (Qi et al., 2017a) and PointNet++ (Qi et al., 2017b) as backbones for point-wise feature extraction. PointRCNN (Shi et al., 2019) is a seminal work in point-based networks, which uses Region Proposal Network (RPN) for 3D proposal generation, followed by a refinement stage, adapted from 2D detection frameworks (Ren et al., 2015). SECOND (Yan et al., 2018) is a pioneering work in voxel-based methods using voxel feature encoders (Zhou & Tuzel, 2018) for efficient point-cloud processing. Following existing work, our work builds upon PointRCNN (Shi et al., 2019) as baseline architectures.

**Unsupervised Domain Adaptation (UDA):** UDA aims to adapt a model trained on labeled source data to perform well on unlabeled target domain data. While 2D vision has seen success with adversarial learning (Chen et al., 2018; Saito et al., 2019) and distribution alignment (Yan et al., 2017), these often require simultaneous access to both source and target data, which renders them ineffective in source-free settings. For source-free settings, self-training has emerged as the dominant UDA strategy, generating pseudo-labels from source-trained models for target fine-tuning. However, existing 3D methods face significant limitations; for example, ST3D (Yang et al., 2021) introduced quality-aware self-training but exhibits inconsistent results. The subsequent improvements, like ST3D++ (Yang et al., 2022), unfortunately, reintroduced source data dependency. These limitations highlight the need for robust, source-free UDA methods in 3D object detection.

**Source-free Domain Adaptation:** Source-free methods (Liang et al., 2020; Saltori et al., 2020; Hegde & Patel, 2024) aims to adapt a source-trained model to an unlabeled target domain without access to the source data. SF-UDA$^{3D}$ (Saltori et al., 2020) is a prominent work, which learns domain-invariant features by disentangling point cloud representations and refining object prototypes using a memory bank. Recent work by Hegde & Patel (2024) improves pseudo-label quality by using a transformer module to filter out low-quality labels.

**Test-Time Domain Adaptation:** Test-time methods enable pre-trained models to adapt during inference without retraining or source data access. Tent (Wang et al., 2020a) represents a seminal general-purpose approach, adapting models by minimizing prediction entropy on target data. The primary existing work is Output Transformation (OT) (Wang et al., 2020b), which applies post-processing corrections by computing dimensional differences between domains and transforming predicted bounding boxes accordingly. This scarcity contrasts with the critical need for robust adaptation in autonomous driving, where deployed models encounter diverse conditions, motivating our development of Test-Time method.

## 3 PROPOSED DOMAIN ADAPTATION METHODS

### 3.1 TEST-TIME STATISTICAL NORMALIZATION (TTSN)

**Test-Time SN (TTSN)** represents our most resource-efficient adaptation method, requiring only two pieces of information: 1) a small quantity (1000 samples in our experiments) of unlabeled target domain data, and 2) statistics about average car size in the target domain – that is, average height ($\mu_h$), width ($\mu_w$), and length ($\mu_l$) of cars in the target domain which can be accessed from the local vehicle registry offices or car-selling websites as discussed by Wang et al. (2020b). Our approach uses the source pretrained model $\theta$ to perform inference on the limited unlabeled target dataset, from which we compute the empirical average of car dimensions ($\bar{y}_h$, $\bar{y}_w$, $\bar{y}_l$). Given the

statistics ($\mu_h$, $\mu_w$, $\mu_l$), we calculate a calibration vector $\boldsymbol{c} = (\Delta y_w, \Delta y_w, \Delta y_l)$ that quantifies the systematic bias in the model's size dimensions predictions:

$$\boldsymbol{c} = (\Delta y_h, \Delta y_w, \Delta y_l) = (\mu_h, \mu_w, \mu_l) - (\overline{y}_h, \overline{y}_w, \overline{y}_l) \tag{1}$$

During test-time, we apply this pre-computed calibration vector to correct the model predictions by adding $\boldsymbol{c}$ to the predicted bounding box dimensions. This post-processing calibration effectively shifts the model's dimensional predictions toward the target domain's car size distribution, thereby mitigating the systematic mislocalization errors that arise from domain-specific size biases.

Note, once the calibration vector is computed, this method then operates entirely at test-time without requiring access to source domain data or fine-tuning the model, making it highly suitable for deployment scenarios with strict computational and source data constraints. Our experimental results demonstrate that TTSN achieves substantial performance improvements, often matching or exceeding the performance of Statistical Normalization (SN) (Wang et al., 2020b), which requires costly model fine-tuning on modified source domain datasets. Algorithm 1 summarizes this method.

---

**Algorithm 1** Test-Time Statistical Normalization (TTSN)

---

**Input:** Source pretrained model $\boldsymbol{\theta_s}$, unlabeled target data $\mathcal{U}_{\text{target}} = \{\boldsymbol{x_i}\}_{i=1}^{N}$ , average car size in the target domain $\boldsymbol{\mu} = (\mu_h, \mu_w, \mu_l)$

**Phase 1: Compute Calibration Factor:**
1: **for** each sample $\boldsymbol{x_i} \in \mathcal{U}_{\text{target}}$ **do**
2: $\quad \widehat{\boldsymbol{y_i}} = \boldsymbol{\theta_s}(\boldsymbol{x_i})$ $\qquad\qquad\qquad\qquad\qquad\qquad\qquad\qquad\qquad\qquad\qquad\quad \triangleright$
$\quad$ where, $\widehat{\boldsymbol{y_i}} = (\widehat{y}_{ih}, \widehat{y}_{iw}, \widehat{y}_{il})$. Note that although the model outputs a bounding box with seven parameters (center coordinates, size dimensions, and rotation), everywhere we are considering only three dimensions just for the purpose of explaining the proposed methods. Also, the model could output multiple bounding boxes, and we would treat each box separately, but the same formulation would carry forward to that case also.
3: **end for**
4: Compute average car dimension $(\overline{y}_h, \overline{y}_w, \overline{y}_l)$ from raw model predictions as follows:

$$\overline{y}_h = \frac{1}{N} \sum_i \widehat{y_{ih}} \;\; ; \quad \overline{y}_w = \frac{1}{N} \sum_i \widehat{y_{iw}} \;\; ; \quad \overline{y}_l = \frac{1}{N} \sum_i \widehat{y_{il}}$$

5: Compute calibration factor: $\mathbf{c} = (\Delta y_h, \Delta y_w, \Delta y_l) = (\mu_h, \mu_w, \mu_l) - (\overline{y}_h, \overline{y}_w, \overline{y}_l)$

**Phase 2: Test-Time Inference:**
1: **for** each test sample $\boldsymbol{x}_{\text{test}}$ **do**
2: $\quad \widehat{\boldsymbol{y}}_{\text{test}} = \boldsymbol{\theta_s}(\boldsymbol{x}_{\text{test}})$
3: $\quad \widetilde{\boldsymbol{y}}_{\text{test}} = \widehat{\boldsymbol{y}}_{\text{test}} + \mathbf{c}$
4: **end for**

**Output:** calibrated predictions $\widetilde{\boldsymbol{y}}_{\text{test}}$

---

## 3.2 Lightweight Linear Scaling (LLS)

**Lightweight Linear Scaling (LLS)** represents our second adaptation approach, specifically designed for scenarios where limited ($\leq 1K$) labeled target domain data is available but the full model fine-tuning is impractical due to insufficient data volume and the risk of catastrophic forgetting of source domain knowledge. This method addresses the fundamental challenge of using small labeled datasets without compromising the pretrained model's learned representations through potentially over-fitted fine-tuning procedures.

In this method, our approach is to introduce a lightweight multiplicative layer that learns domain-specific scaling transformations to align raw predictions with the target domain's car size dimensions. Specifically, we learn three scale factors $\boldsymbol{s} = (s_h, s_w, s_l)$ that transform the raw car size predictions $(\widehat{y}_h, \widehat{y}_w, \widehat{y}_l)$ from the source model to produce calibrated final predictions $(\widetilde{y}_h, \widetilde{y}_w, \widetilde{y}_l)$ through element-wise multiplication: $\widetilde{y}_h = \widehat{y}_h \cdot s_h, \qquad \widetilde{y}_w = \widehat{y}_w \cdot s_w, \qquad \widetilde{y}_l = \widehat{y}_l \cdot s_l$.

The multiplicative layer is integrated as the final transformation stage in the detection pipeline, requiring only three additional parameters while preserving all pretrained model weights. To optimize these scale factors, we employ a three-stage training protocol – 1) Raw Prediction Generation: We apply the source model $\theta_s$ to generate initial predictions on the available labeled target domain samples; 2) High-Quality Data Curation: We construct a refined training subset by retaining only predictions with Intersection over Union (IoU) values exceeding 0.5 between predicted and ground-truth bounding boxes, ensuring training stability and reducing the impact of low-quality predictions; and 3) Scale Factor Optimization: We train a lightweight linear regression model to learn the optimal scale factors using the curated high-quality prediction-ground truth pairs.

This approach (Algorithm 2) effectively decouples dimensional calibration from feature learning, allowing the model to adapt to target domain size distributions while maintaining the rich semantic representations acquired during source domain pre-training.

---

**Algorithm 2** Lightweight Linear Scaling (LLS)

---

    **Input:** A pretrained source model $\theta_s$ and a labeled target domain dataset $\mathcal{L}_{\text{target}} = \{(x_i, y_i)\}_{i=1}^N$.

    **Phase 1: High-Quality Data Curation:**
1:  Initialize an empty list of high-quality predictions: $\mathcal{P}_{\text{HQ}} = \emptyset$
2:  **for** each labeled sample $(x_i, y_i) \in \mathcal{L}_{\text{target}}$ **do**       ▷ where, $y_i = (y_{ih}, y_{iw}, y_{il})$
3:      Get raw predictions from the source model: $\widehat{y_i} = \theta_s(x_i)$   ▷ where, $\widehat{y_i} = (\widehat{y_{ih}}, \widehat{y_{iw}}, \widehat{y_{il}})$
4:      Evaluate IoU between $\widehat{y_i}$ and ground truth $y_i$.
5:      **if** IoU$(\widehat{y_i}, y_i) > 0.5$ **then**
6:         Add pair to list: $\mathcal{P}_{\text{HQ}} = \mathcal{P}_{\text{HQ}} \cup \{(\widehat{y_i}, y_i)\}$
7:      **end if**
8:  **end for**

    **Phase 2: Scale Factor Optimization:**
1:  Perform linear regression to learn scale factors $s = (s_h, s_w, s_l)$
2:  Minimize the loss function and store the optimized scale factors $s^*$:

$$s^* = \arg\min_s \mathcal{L}(s) = \arg\min_s \frac{1}{M} \sum_{(\widehat{y_i}, y_i) \in \mathcal{P}_{\text{HQ}}} \|y_i - \widehat{y_i} \odot s\|_2^2 \tag{2}$$

     ▷ where, $M$ is number of pairs in $\mathcal{P}_{\text{HQ}}$ and $\odot$ is element wise multiplication of two vectors.

    **Phase 3: Test-Time Inference:**
1:  For a given test image $x_{\text{test}}$:
2:  Get raw predictions from the source model: $\widehat{y}_{\text{test}} = \theta_s(x_{\text{test}})$
3:  Apply the optimized scale factors: $\widetilde{y}_{\text{test}} = \widehat{y}_{\text{test}} \odot s^*$

    **Output:** Calibrated predictions $\widetilde{y}_{\text{test}}$ for the target domain.

---

## 3.3 Weakly Supervised Fine-tuning (WSFT)

Weakly-Supervised Fine-tuning (WSFT) constitutes our third adaptation strategy, designed for scenarios where source domain data remains inaccessible but a substantial volume of unlabeled target domain data is available alongside the car size statistics. Given the prohibitive cost of point cloud annotation, this approach enables adaptation of source pretrained models without requiring human labeling. Building upon prior work demonstrating the usefulness of unsupervised fine-tuning for mitigating source domain biases (Saltori et al., 2020; Yang et al., 2021; Xu et al., 2021), our method specifically addresses dimensional misalignment, which is one of the most detrimental forms of domain shift in 3D object detection. In what follows, we describe the WSFT approach.

Let $\theta_s$ be the source trained model and $\mathcal{U}_{\text{target}} = \{x_i\}_{i=1}^N$ be a set of unlabeled target domain data. Further, let $\mu = (\mu_h, \mu_w, \mu_l)$ be the average car size statistics in the target domain. The WSFT method aims to minimize the following *Composite Statistical Alignment (CSA)* loss function and

Table 1: Dataset overview

| Dataset | Train Size | Val Size | Regions | Classes |
|---------|-----------:|---------:|---------|--------:|
| KITTI | 3,712 | 3,769 | Karlsruhe, Germany | 8 |
| Waymo | 26,000 | 6,000 | Phoenix, Mountain View, San Francisco | 4 |
| nuScenes | 8,614 | 1,319 | Boston and Singapore | 23 |
| Argoverse | 12,756 | 3,817 | Miami and Pittsburgh | 17 |

get closer to the optimal value $\boldsymbol{\theta}_{\text{WSFT}}$.

$$\boldsymbol{\theta}_{\text{WSFT}} = \underset{\boldsymbol{\theta}}{\arg\min} \, \mathcal{L}(\boldsymbol{\theta}) = \frac{1}{N} \sum_{i=1}^{N} \left[ (1-\alpha) \, \|\overline{\boldsymbol{y}}(\boldsymbol{\theta}) - \boldsymbol{\mu}\|_2^2 + \alpha \, \|\widehat{\boldsymbol{y}}_i(\boldsymbol{\theta}) - \widehat{\boldsymbol{y}}_i(\boldsymbol{\theta_s})\|_2^2 \right] \quad (3)$$

where, $\widehat{\boldsymbol{y}}_i(\boldsymbol{\theta_s}) = \boldsymbol{\theta_s}(\boldsymbol{x_i})$ is the prediction of the source model $\boldsymbol{\theta_s}$ on example $\boldsymbol{x_i}$. Similarly, $\widehat{\boldsymbol{y}}_i(\boldsymbol{\theta})$ is the the prediction of the model $\boldsymbol{\theta}$ on the same example $\boldsymbol{x_i}$. The quantity $\overline{\boldsymbol{y}}(\boldsymbol{\theta}) = \frac{1}{N} \sum_i \widehat{\boldsymbol{y}}_i(\boldsymbol{\theta})$ is the average car size statistic as per model $\boldsymbol{\theta}$.

The above CSA loss comprises two complementary terms: 1) *statistical alignment term* (weighted by $1-\alpha$) minimizes the deviation between the model-based empirical average of car size dimensions in the target domain versus true statistics, encouraging the distribution of model predictions to align with the target domain's car size distribution, and 2) *consistency regularization term* (weighted by $\alpha$) penalizes deviation of learned model $\boldsymbol{\theta}$'s predictions from the corresponding reference predictions obtained through source model $\boldsymbol{\theta_s}$, preventing the learning process from converging to trivial solutions where all predictions $\widehat{\boldsymbol{y}}_i(\boldsymbol{\theta})$ equal $\boldsymbol{\mu}$. This dual-objective formulation ensures that dimensional alignment occurs through meaningful adaptation rather than degenerate prediction collapse.

We solve the above optimization problem via Stochastic Gradient Descent (SGD) Algorithm (Ruder, 2016). At any $k^{th}$ step during the run of the SGD algorithm, the above CSA loss function gets confined only to the examples present in the corresponding batch $\mathcal{B}_k \subset \mathcal{U}_{\text{target}}$. In this case, the quantity $\overline{\boldsymbol{y}}(\boldsymbol{\theta})$ is computed over all the examples in the batch $\mathcal{B}_k$ and $\boldsymbol{\theta}$ corresponds to the value of the learned parameter till that time, say $\boldsymbol{\theta}_k$. To ensure high-quality pseudo-labels, we filter $\widehat{\boldsymbol{y}}_i(\boldsymbol{\theta_s})$ based on model confidence scores. Specifically, we use the logit outputs as confidence estimates and apply a threshold of 1 to discard low-confidence predictions prior to greedy matching. See Algorithm 3 for an algorithmic description.

Our experiments show substantial performance improvements of $40 - 50\%$ compared to unadapted baselines, validating both the effectiveness of our composite loss formulation and the overall WSFT framework for addressing dimensional misalignment in cross-domain 3D object detection.

## 4 EXPERIMENTS

We evaluate our domain adaptation methods on datasets from Germany, USA, and Singapore. Our implementation utilizes code from (Wang et al., 2020b) for converting datasets to the standard KITTI format and evaluation. We use PyTorch framework (Paszke et al., 2019) for training.

**Datasets:** We demonstrate the robustness of our methods across four large-scale autonomous driving datasets spanning diverse geographical regions. The KITTI dataset (Geiger et al., 2012; 2013) serves as our primary benchmark, containing 7481 annotated LiDAR frames captured in Karlsruhe, Germany. The Waymo Open dataset (Sun et al., 2020) provides the largest evaluation corpus. From this dataset, we subsample 26000 training and 6000 validation frames for multi-city scenes (Phoenix, Mountain View, San Francisco) encompassing varied weather and lighting conditions. The nuScenes dataset (Caesar et al., 2020) captures urban environments across Boston and Singapore with 32-beam LiDAR. For this dataset, we re-split its training set into 8614 training samples and 1319 validation samples. The Argoverse dataset (Chang et al., 2019) represents Miami and Pittsburgh driving scenarios. We subsample 12756 (3817) frames from training (validation) splits. See Table 1 for details.

For our TTSN method, we use a calibration subset of 1000 unlabeled samples from each target domain dataset to compute the calibration factor $c$. This sample size choice was based on the argument that using smaller subsets may introduce excessive noise in the dimensional average estimation, while larger subsets compromise the resource-constrained deployment scenario that motivates

Table 2: TTSN is robust to deterioration unlike OT (Wang et al., 2020b) (KITTI → nuScenes)

| Method | $AP_{BEV}$ | | | $AP_{3D}$ | | |
|---|---|---|---|---|---|---|
| | Easy | Moderate | Hard | Easy | Moderate | Hard |
| DT | 47.3 | 24.6 | 25.4 | 14.3 | 8.6 | 8.8 |
| OT | 55.0 | 30.8 | 27.8 | 10.4 | 6.8 | 7.6 |
| TTSN (Ours) | 67.9 | 34.7 | 30.7 | 25.4 | 12.8 | 16.1 |
| Target | 73.4 | 40.7 | 40.2 | 38.1 | 21.2 | 20.5 |
| Gap Closed : OT | 29.5% | 38.6% | 16.2% | -16.3% | -14.4% | -10.2% |
| Gap Closed : TTSN | **78.9%** | **62.8%** | **35.8%** | **46.6%** | **33.4%** | **62.3%** |

our approach. (See section A.2 for set-size ablation) . Following (Wang et al., 2020b), for experimentation purpose, we obtain the target statistics $\boldsymbol{\mu} = (\mu_h, \mu_w, \mu_l)$ directly from the dataset by computing averages (See section A.1 for results using car sales data). Note, prior work has demonstrated that domain adaptation methods still remain effective when target statistics are derived from publicly available car sales databases rather than dataset-specific computations (Wang et al., 2020b), indicating the practical feasibility of our approach.

**Models:** We employ PointRCNN (Shi et al., 2019) as our base 3D object detection framework to validate the effectiveness of our proposed domain adaptation methods. PointRCNN processes raw LiDAR point clouds through a PointNet++ (Qi et al., 2017b) backbone for point-wise feature extraction, operating entirely on 3D point-cloud data without auxiliary image modalities. PointRCNN uses a two-stage detection framework, where each stage is trained separately. In the first stage, the RPN sub-network generates coarse 3D bounding box proposals from point features. In the second stage, the RCNN sub-network refines these proposals for better localization and classification. We utilize the pretrained checkpoints provided by Wang et al. (2020b), trained on each source domain dataset. The pre-training procedure involved training the RPN stage for 200 epochs using a batch size of 16 and a learning rate of 0.02, followed by training the RCNN stage refinement for 70 epochs using a batch size of 4 and a learning rate of 0.02. We also verify the generalizability of our methods on another 3D detection method, SECOND-iou (Yan et al., 2018), results for which are presented in the section A.3.

**Metrics:** We evaluate all methods using *Average Precision (AP)* computed in both *Bird's Eye View (BEV)* and *3D*, denoted as $AP_{BEV}$ and $AP_{3D}$, respectively. Following existing work, we focus primarily on the *Car* class, which represents the most prevalent class in autonomous driving datasets. For both AP scores, we use an Intersection over Union (IoU) threshold of 0.7 to classify predictions as true positives.

**Optimization:** For LLS, we optimize the dimensional scale factors $(s_h, s_w, s_l)$ using the Adam optimizer (Kingma, 2014) with $LR = 0.01$. For WSFT, we perform end-to-end fine-tuning of all parameters in the source model $\theta$ using the Adam optimizer and a value of $\alpha = 0.1$ in the loss function. We use a batch size of 8 and a reduced $LR = 7.8125e - 6$ to reduce memory usage and support training on a single Tesla V100 GPU.

**Baselines:** We compare our methods against approaches using the source data, those using labeled target data, and test-time adaptation methods:

- *Direct Transfer (DT)* applies a source pretrained detector to the target domain without adaptation,

- *Statistical Normalization (SN)* (Wang et al., 2020b) modifies the source dataset's point clouds and labels by adding the mean car sizes difference between target domain and source domain, and then retrains the source model on this dataset,

- *Output Transformation (OT)* (Wang et al., 2020b) directly adjusts the detector's prediction by adding the mean car size difference between target and source domain to the predicted size,

- *Few-Shot Fine-Tuning (FS)* (Wang et al., 2020b) fine-tunes the pretrained object detector with 10 labeled instances of the target domain,

Table 3: Domain adaptation results using a KITTI source-trained PointRCNN

| | KITTI → nuScenes | | | | | | KITTI → Waymo | | | | | |
|---|---|---|---|---|---|---|---|---|---|---|---|---|
| | $AP_{BEV}$ | | | $AP_{3D}$ | | | $AP_{BEV}$ | | | $AP_{3D}$ | | |
| **Method** | Easy | Moderate | Hard | Easy | Moderate | Hard | Easy | Moderate | Hard | Easy | Moderate | Hard |
| DT | 47.3 | 24.6 | 25.4 | 14.3 | 8.6 | 8.8 | 41.0 | 35.6 | 36.8 | 10.4 | 10.6 | 10.4 |
| SN | 60.8 | 32.9 | **31.9** | 23.9 | **16.4** | 15.8 | 84.6 | 74.9 | 69.4 | 53.3 | 49.4 | 49.4 |
| OT | 55.0 | 30.8 | 27.8 | 10.4 | 6.8 | 7.6 | 86.1 | 69.1 | 68.7 | 16.2 | 13.1 | 13.9 |
| FS | 54.7 | 28.7 | 27.5 | 21.7 | 12.5 | 12.4 | 87.4 | **75.9** | 70.1 | 70.9 | **55.3** | **54.4** |
| TTSN (ours) | **67.9** | **34.7** | 30.7 | 25.4 | 12.8 | 16.1 | 89.5 | 75.8 | 70.7 | 65.8 | 46.9 | 44.4 |
| LLS (ours) | 67.8 | 34.6 | 30.6 | **29.4** | 16.1 | **16.5** | 89.4 | 75.8 | **70.8** | 72.1 | 54.5 | 52.6 |
| WSFT (ours) | 59.3 | 31.1 | 28.4 | 19.6 | 10.4 | 10.6 | **89.8** | 75.0 | 70.3 | **72.1** | 54.4 | 53.0 |
| Target | 73.4 | 40.7 | 40.2 | 38.1 | 21.2 | 20.5 | 90.1 | 85.9 | 80.4 | 85.3 | 67.9 | 67.7 |

- *Target* corresponds to a model that is trained from scratch on the labeled data of the target domain. This is a possible upper bound of performance after adaptation. (Wang et al., 2020b)

## 4.1 RESULT ANALYSIS

**TTSN achieves a new state-of-the-art for test-time adaptation:** Table 2 shows adaptation effectiveness through gap closure percentage, given by $\frac{AP_{\text{adapted}} - AP_{\text{DT}}}{AP_{\text{target}} - AP_{\text{DT}}} \times 100$, measuring the fraction of domain-shift performance loss recovered by each method. We see that TTSN consistently outperforms OT – the only existing test-time method to our knowledge – across all evaluation scenarios. Notably, while OT suffers from performance degradation in 3D evaluation (declining by 16.3%, 14.4%, and 10.2% for easy, moderate, and hard cases, respectively), TTSN achieves substantial improvements of 46.6%, 33.4%, and 62.3%, respectively. This establishes TTSN as the first test-time adaptation approach delivering consistent performance gains without sacrificing detection accuracy.

**TTSN outperforms with less data and computation:** TTSN uses only $1K$ unlabeled samples from the target-domain and requires no training, but still outperforms competitor methods that require substantially more data or computational resources.

In the *KITTI → Waymo* domain shift scenario (Table 3), our TTSN surpasses SN, which requires source data access and model finetuning, achieving AP scores of 89.5, 75.8, 70.7 versus SN's 84.6, 74.9, 69.4 across easy, moderate, and hard cases in bird's eye view. TTSN also outperforms Few-Shot fine-tuning (FS), which demands labeled target data.

For *KITTI → nuScenes* domain shift scenario (Table 3) reinforces this trend, with TTSN consistently outperforming SN, OT, and FS. Notably, TTSN achieves these improvements while operating under the most restrictive resource constraints, requiring neither source data access, labeled target samples, nor fine-tuning. For additional *KITTI → Argoverse* results, see Table 5 in the Appendix.

**WSFT and LLS achieve better results in 3D:** While TTSN performs on par with LLS and WSFT in terms of $AP_{BEV}$ score, Table 3 show that both methods outperform TTSN in the $AP_{3D}$ scores. We argue this is due to fundamental differences in adaptation mechanisms. LLS uses true labels to learn optimal scale factors, enabling precise dimensional corrections compared to TTSN's reliance on noisy pseudo-labels from raw source predictions. WSFT performs full model finetuning on a larger unlabeled dataset, allowing comprehensive feature space adjustments that improves 3D geometric reasoning. In contrast, TTSN does simple post-processing adjustment without modifying learned representations, limiting its ability to address complex 3D localization errors.

**LLS outperforms FS without altering trained parameters:** LLS corrects dimensional errors by learning three scale factors through independently training a linear regression layer on labeled target data, leaving all source detector parameters unchanged. In contrast, Few-Shot fine-tuning (FS) uses the same labeled target data but updates all source-trained parameters. Counterintuitively, despite FS's access to the full parameter space for adaptation, LLS demonstrates consistent superiority across both domain shifts (Table 3). In KITTI → nuScenes adaptation, LLS substantially outperforms FS with $AP_{BEV}$ scores of 67.8, 34.6, 30.6 versus FS's 54.7, 28.7, 27.5, and $AP_{3D}$ scores of 29.4, 16.1, 16.5 vs 21.7, 12.5, 12.4 across easy, moderate, and hard categories, respectively.

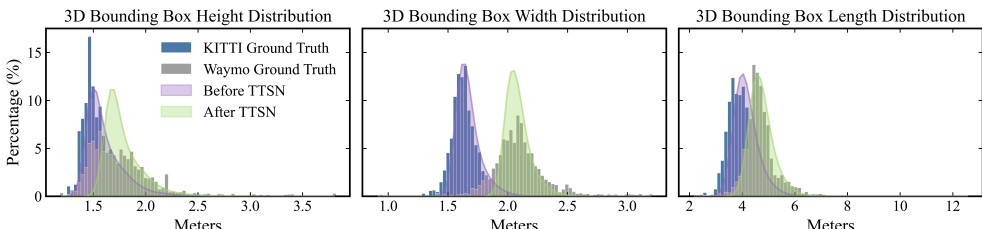

Figure 3: Size-distribution of predictions from a KITTI-trained detector on Waymo shifted towards Waymo ground truth distribution after applying TTSN

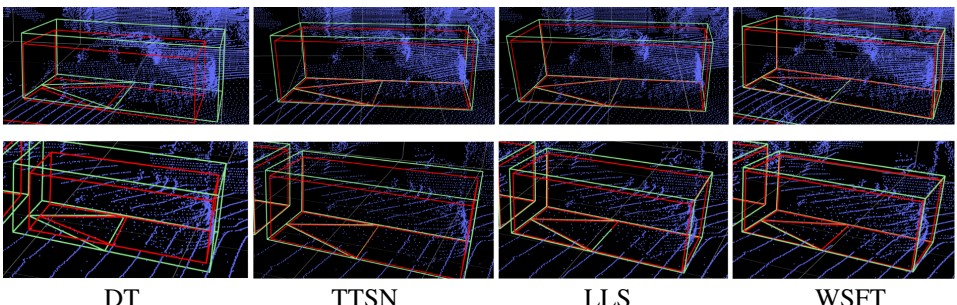

Figure 4: Our domain adaptation methods align predicted bounding boxes with ground truth dimensions. Green boxes indicate ground truth, while red boxes show predictions.

**All three methods adapt to the target without source:** Unlike most existing domain adaptation approaches (Wang et al., 2023; Yang et al., 2021; 2022; Wang et al., 2020b; Zhang et al., 2021), our methods operate under source-data-free constraints, addressing practical deployment scenarios where pre-trained models are publicly available but training datasets remain proprietary due to substantial annotation costs of 3D data. Our results in Table 3 demonstrates that our source-free adaptations perform comparable to or better than source-dependent methods, establishing the practical viability of effective domain adaptation without requiring access to original training data.

**TTSN realigns prediction distributions toward target domain:** Figure 3 demonstrates dimensional distribution shifts across height, width, and length dimensions following TTSN application. The pre-adaptation distribution closely aligns with KITTI ground-truth characteristics, confirming the presence of source domain dimensional bias in the pre-trained detector, as previously identified by Wang et al. (2020b). After applying TTSN, the adapted predictions move much closer to the Waymo ground truth, showing that TTSN effectively fixes cross-domain dimensional misalignment. See Figure 4 for visual results.

**TTSN, LLS, and WSFT outperform competition across datasets:** We evaluate our methods across multiple source-target domain pairs using KITTI as the source domain and Waymo, nuScenes, and Argoverse as target domains (see Tables 3 and 5). These datasets encompass diverse geographical contexts and sensor configurations, with Waymo employing 64-beam LiDAR while nuScenes utilizes 32-beam sensors, creating both environmental and hardware-induced domain shifts. Our methods consistently outperform unadapted baselines and existing approaches across all scenarios, showcasing their robustness in addressing geographical variations, environmental conditions, and LiDAR sensor differences, making them robust solutions for cross-domain 3D object detection.

## 5 ABLATIONS

### 5.1 IMPORTANCE OF CONSISTENCY REGULARIZATION TERM IN CSA LOSS

The proposed CSA loss for WSFT method comprises two components: the *statistical alignment term*, which aligns model predictions with the target domain's car size distribution, and the *consistency regularization term*, which acts as a regularizer to prevent predictions from overfitting to

the average target car size. Table 4 presents an ablation study assessing the impact of this regularization. We observe that without the consistency term, the model overfits to the target domain's mean statistics. Although this results in a lower training loss, it causes a sharp decline in validation performance; notably, the model collapses after only three epochs, degenerating to a state where it predicts only the mean dimension values.

Table 4: WSFT without consistency term in loss function (KITTI → Waymo)

| Epoch no. | $AP_{BEV}$ | | | $AP_{3D}$ | | |
| | Easy | Moderate | Hard | Easy | Moderate | Hard |
|---|---|---|---|---|---|---|
| DT | 41.0 | 35.6 | 36.8 | 10.4 | 10.6 | 10.4 |
| Epoch 1 | 81.8 | 68.8 | 65.3 | 40.1 | 33.8 | 35.4 |
| Epoch 2 | 33.9 | 33.7 | 35.5 | 6.5 | 6.9 | 13.6 |
| Epoch 3 | 6.9 | 8.0 | 10.3 | 0.3 | 1.0 | 1.0 |
| Consistency on | **89.8** | **75.0** | **70.3** | **72.1** | **54.4** | **53.0** |

## 6 CONCLUSION

This work addresses the critical challenge of domain shift in LiDAR-based 3D object detection, where models trained in one geographic location suffer significant performance drops when deployed in another. We introduced a suite of three practical, source-free domain adaptation methods, called as TTSN, LLS, and WSFT – each tailored to different resource and data availability scenarios. Extensive experiments demonstrate that the proposed methods effectively mitigate performance degradation by correcting for dimensional biases between domains. The proposed methods consistently outperform existing techniques across diverse datasets, including Waymo, nuScenes, and Argoverse.

### ETHICAL STATEMENT

This research is built upon open-source datasets and leverages a pre-trained checkpoint from a foundational work (Wang et al., 2020b). Our methodology is designed to be transparent and reproducible, relying exclusively on publicly available resources. While the proposed methods show significant improvement over the current state-of-the-art, they still fall short of ideal scores needed for reliable deployment in real-world scenarios. Real-world deployment requires careful consideration, such as integration of these perception modules with other downstream components of the autonomous vehicle system. A thorough evaluation in a simulated environment must be performed with the entire integrated pipeline to avoid any mishap.

### REPRODUCIBILITY STATEMENT

To ensure reproducibility, we have provided detailed methodological descriptions and algorithmic pseudo-code for all proposed approaches. All experimental hyperparameters and training configurations are clearly articulated throughout the manuscript. The complete source code will be made publicly accessible post-acceptance to promote replication and advance future research in this field.

### USE OF LARGE LANGUAGE MODELS

In accordance with ICLR 2026 policy, we would like to disclose that this manuscript was prepared with the assistance of Large Language Models (LLMs), specifically Gemini 2.5 Pro. As part of our commitment to transparency, we want to clarify that these tools were utilized solely for editorial and organizational purposes. Their role was restricted to enhancing the clarity, readability, and overall flow of certain sections, ensuring effective communication of the research.

The intellectual foundation of this study, including the conception of ideas, experimental design, and analysis of results, is entirely the work of the authors. The LLM's involvement was limited

to language refinement and did not influence the scientific content or conclusions presented. The authors have thoroughly reviewed and validated all aspects of this work and take full responsibility for the final submission.

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

# A APPENDIX

Table 5: Domain adaptation results for KITTI (source) → Argoverse (target)

| Method | $AP_{BEV}$ Easy | Moderate | Hard | $AP_{3D}$ Easy | Moderate | Hard |
|---|---|---|---|---|---|---|
| DT | 50.6 | 41.5 | 40.8 | 22.6 | 18.3 | 20.3 |
| SN | 74.7 | **61.5** | 60.6 | 48.2 | **38.2** | **37.1** |
| OT | 72.7 | 59.9 | 59.3 | 9 | 7.9 | 9.3 |
| FS | 75.8 | 60.7 | 59.8 | **49.2** | 37.3 | 36.5 |
| TTSN (ours) | 73.7 | 60.5 | **60.7** | 35.6 | 24.4 | 25.5 |
| LLS (ours) | 72.9 | 59.9 | 59.9 | 41.2 | 30.4 | 30.5 |
| WSFT (ours) | **75.9** | 60.2 | 56.3 | 46.0 | 34.0 | 33.5 |
| Target | 79.2 | 69.9 | 69.9 | 57.8 | 44.2 | 42.8 |

---

**Algorithm 3** Weakly Supervised Fine-tuning (WSFT)

---

**Input:** A pre-trained source model with parameters $\boldsymbol{\theta}$, a large unlabeled target domain dataset $\mathcal{U}_{\text{target}}$, and average car size in the target domain $\boldsymbol{\mu} = (\mu_h, \mu_w, \mu_l)$

**Phase 1: Pseudo-label Generation**
1: Create an empty list for reference predictions: $\mathcal{R} = \emptyset$.
2: **for** each sample $\boldsymbol{x_i} \in \mathcal{U}_{\text{target}}$ **do**
3:     Get predictions from the frozen source model: $\widehat{\boldsymbol{y_i}}(\boldsymbol{\theta}) = \boldsymbol{\theta}(\boldsymbol{x_i})$
4:     Add predictions to the reference list: $\mathcal{R} = \mathcal{R} \cup \{\widehat{\boldsymbol{y_i}}(\boldsymbol{\theta})\}$
5: **end for**

**Phase 2: Statistical Alignment Training**
1: Initialize training loop. Set $\boldsymbol{\theta_0} \leftarrow \boldsymbol{\theta}$
2: **for** $k = 1$ to $K$ training steps **do**
3:     $\mathcal{M}_k \leftarrow \emptyset$
4:     Sample a mini-batch of data $\mathcal{B}_k = \{\boldsymbol{x_1}, \boldsymbol{x_2}, \ldots, \boldsymbol{x_m}\}$ from $\mathcal{U}_{\text{target}}$.
5:     **for** each example $\boldsymbol{x_j} \in \mathcal{B}_k$ **do**
6:         Get predictions from the current model $\boldsymbol{\theta_{k-1}}$ as follows: $\widehat{\boldsymbol{y_j}}(\boldsymbol{\theta_{k-1}}) = \boldsymbol{\theta_{k-1}}(\boldsymbol{x_j})$
7:         Retrieve corresponding reference predictions $\widehat{\boldsymbol{y_j}}(\boldsymbol{\theta_0})$ from $\mathcal{R}$.
8:         Use 3D IoU score and greedy matching to check if $\widehat{\boldsymbol{y_j}}(\boldsymbol{\theta_{k-1}})$ matches with $\widehat{\boldsymbol{y_j}}(\boldsymbol{\theta_0})$.
9:         **if** matching is successful **then**
10:             $\mathcal{M}_k \leftarrow \mathcal{M}_k \cup \{\widehat{\boldsymbol{y_j}}(\boldsymbol{\theta_{k-1}})\}$
11:         **end if**
12:     **end for**
13:     **if** $\mathcal{M}_k$ is not empty **then**
14:         Compute batch-averaged current predictions: $\overline{\boldsymbol{y}}(\boldsymbol{\theta_{k-1}}) = \frac{1}{|\mathcal{M}_k|} \sum_{\mathcal{M}_k} \widehat{\boldsymbol{y_j}}(\boldsymbol{\theta_{k-1}})$
15:         Compute the composite statistical alignment loss:

$$\mathcal{L} = \frac{1}{|\mathcal{M}_k|} \sum_{\mathcal{M}_k} \left[ (1 - \alpha) \left\| \overline{\boldsymbol{y}}(\boldsymbol{\theta_{k-1}}) - \boldsymbol{\mu} \right\|_2^2 + \alpha \left\| \widehat{\boldsymbol{y_j}}(\boldsymbol{\theta_{k-1}}) - \widehat{\boldsymbol{y_j}}(\boldsymbol{\theta_0}) \right\|_2^2 \right]$$

16:         Backpropagate and update model parameters and call that updated parameter as $\boldsymbol{\theta_k}$.
17:     **end if**
18: **end for**

**Output:** Fine-tuned model $\boldsymbol{\theta_K}$ adapted to the target domain.

---

Table 6: TTSN using the mean sizes of datasets versus car sales data (KITTI → Waymo)

| | $AP_{BEV}$ | | | $AP_{3D}$ | | |
|---|---|---|---|---|---|---|
| **Source** | Easy | Moderate | Hard | Easy | Moderate | Hard |
| DT | 41.0 | 35.6 | 36.8 | 10.4 | 10.6 | 10.4 |
| Dataset | 89.5 | 75.8 | 70.7 | 65.8 | 46.9 | 44.4 |
| Car sales data | 89.4 | 75.7 | 70.7 | 65.8 | 46.8 | 44.3 |

## A.1 SENSITIVITY TO ERRORS IN TARGET STATISTICS

In the main paper, we compute target statistics ($\mu$) as the average car size in within each dataset. Additionally, we conducted an ablation study to evaluate TTSN's performance when using statistics derived from real-world car sales data. For this, we utilize the average car sizes collected by Wang et al. (2020b), which cite the average car size in the USA as $(1.75, 1.93, 5.15)$.

Table 6 presents the results of the KITTI → Waymo domain adaptation task, comparing the use of sales data against dataset statistics. We observe that performance remains largely unchanged regardless of the source of the statistics.

Additionally, we also conducted a study around how the performance of TTSN would vary based on the error rate in target statistics computation. Table 7 shows the effect of both overestimation and underestimation in $\mu$.

We observe that the performance drops linearly when the mean car size statistics are overestimated. Interestingly, a slight underestimation of these statistics leads to a performance improvement, but a large underestimation causes a performance drop. We attribute this behavior to the bias-variance tradeoff. Pseudo labels (predictions of the source model on the target domain data) consist of a consistent bias and random noise. Our mean size correction aims to eliminate the bias, but this process also increases the noise. A small underestimation subtly reduces the correction, which still removes most of the bias while injecting less noise, thus improving performance. Conversely, overestimation always overshoots the true value, worsening both the bias and the noise. A significant underestimation leaves too much bias uncorrected, which is why performance decreases in that scenario. This observed trend reflects the balance between effective bias removal and minimal noise amplification.

Table 7: TTSN Sensitivity to errors in target domain car size statistics (KITTI → Waymo)

| | Under-estimation | | | | | | Over-estimation | | | | | |
|---|---|---|---|---|---|---|---|---|---|---|---|---|
| | $AP_{BEV}$ | | | $AP_{3D}$ | | | $AP_{BEV}$ | | | $AP_{3D}$ | | |
| **Error** | Easy | Moderate | Hard | Easy | Moderate | Hard | Easy | Moderate | Hard | Easy | Moderate | Hard |
| 0% | 89.5 | 75.8 | 70.7 | 65.8 | 46.9 | 44.4 | **89.5** | **75.8** | **70.7** | **65.8** | **46.9** | **44.4** |
| 1% | 89.4 | 75.8 | 70.8 | 70.0 | 50.7 | 46.8 | 89.4 | 75.6 | 70.6 | 59.9 | 40.6 | 39.8 |
| 3% | **89.5** | **75.9** | **70.8** | 72.9 | 54.7 | 52.7 | 88.7 | 74.8 | 69.9 | 37.0 | 27.2 | 28.3 |
| 5% | 89.2 | 75.7 | 70.6 | 72.9 | **55.2** | **53.1** | 87.1 | 68.7 | 68.4 | 20.8 | 15.2 | 16.6 |
| 8% | 88.0 | 70.1 | 69.5 | 66.0 | 50.4 | 46.2 | 79.9 | 63.4 | 62.5 | 5.5 | 4.6 | 5.6 |
| 10% | 84.5 | 68.5 | 67.1 | 49.8 | 38.9 | 38.2 | 65.9 | 55.2 | 52.7 | 1.5 | 1.9 | 2.2 |
| Target | 73.4 | 40.7 | 40.2 | 38.1 | 21.2 | 20.5 | 90.1 | 85.9 | 80.4 | 85.3 | 67.9 | 67.7 |

## A.2 VARYING CALIBRATION SET SIZES

In the primary analysis, we utilized a calibration set size of $1k$, To investigate the sensitivity of TTSN to data volume and determine if data requirements could be further reduced without compromising accuracy, we present an ablation study on set sizes of $200, 500, 1k$, and $2k$ in Table 8 We observe that detection quality remains largely consistent across set sizes of $500$ and above. While the $200$-sample set exhibits a slight decline in 3D scores, the $1k$ set size achieves optimal performance, thereby validating its selection as the standard for our experiments.

Table 8: TTSN using varying size of the calibration set (KITTI → Waymo)

| | $AP_{BEV}$ | | | $AP_{3D}$ | | |
|---|---|---|---|---|---|---|
| **Calibration set size** | Easy | Moderate | Hard | Easy | Moderate | Hard |
| 200 | 89.5 | 75.8 | 70.7 | 64.6 | 43.4 | 43.8 |
| 500 | 89.5 | 75.8 | 70.7 | 65.7 | 46.8 | 44.3 |
| 1000 | 89.5 | 75.8 | 70.7 | **65.8** | **46.9** | **44.4** |
| 2000 | 89.5 | 75.8 | 70.7 | 65.9 | 46.9 | 44.4 |

## A.3 SECOND-IOU RESULTS

In Table 9, we present object detection performance utilizing the SECOND-IoU framework (Yan et al., 2018). This architecture is a single-stage, voxel-based method that employs 3D sparse convolutions coupled with a Region Proposal Network (RPN) for object localization and classification. Adopting the implementation and pretrained checkpoints from Hegde & Patel (2024), these experiments demonstrate the effective generalization of our approach. The consistent improvements observed here verify that TTSN and LLS are model-agnostic and robust across different detection backbones.

Table 9: Results on SECOND-iou (Waymo → KITTI)

| | $AP_{BEV}$ | | | $AP_{3D}$ | | |
|---|---|---|---|---|---|---|
| **Methods** | Easy | Moderate | Hard | Easy | Moderate | Hard |
| DT | 86.8 | 66.5 | 65.8 | 54.4 | 40.8 | 39.8 |
| OT | 11.0 | 11.7 | 12.7 | 0.1 | 0.1 | 0.1 |
| TTSN | **89.1** | **68.5** | **68.1** | 70.9 | 47.7 | 46.8 |
| LLS | 88.8 | 68.2 | 68.0 | **73.3** | **52.3** | **48.6** |

