# OpenReview forum: "Training & Label-Free Domain Adaptation in 3D Object Detection for Autonomous Driving"
_ICLR.cc/2026/Conference — Submitted to ICLR 2026_

### Official Review · Reviewer_e7Me · 2025-10-31

**Soundness:** 3
**Presentation:** 3
**Contribution:** 3
**Rating:** 6
**Confidence:** 4

**Summary:**

This paper provides a detailed review of the research status of domain adaptation in the field of 3D object detection over recent years, mainly addressing the issue of detector performance degradation caused by differences in vehicle size between the source and target domains. It proposes three domain adaptation methods suitable for different scenarios. TTSN uses only a small amount of unlabeled target domain data and vehicle size statistics information to calibrate the prediction boxes during the test inference phase. LLS uses only a small amount of labeled target domain data and calibrates the prediction boxes by training a lightweight scaling factor without changing the model parameters. WSFT fine-tunes the model using a large amount of unlabeled target domain data through the CSA loss function, thereby enhancing its generalization ability.

**Strengths:**

1.The three proposed methods cover different scenarios and offer high flexibility in practical applications.

2.They address the issue of performance degradation caused by differences in vehicle size between different domains.

3.A large number of experiments have been conducted to prove the effectiveness of the proposed three methods.

**Weaknesses:**

1.The study only focuses on vehicle size deviations and does not investigate other causes that may lead to a decrease in detector performance, such as point cloud density differences.

2.It only studies the vihicle task and has not verified whether it remains effective for categories with less obvious size differences, such as pedestrians and cyclists.

**Questions:**

Is there enough innovation? As I understand, the TTSN method is just applying the SN method to the TTA task. And how effective are these three proposed methods for pedestrians and cyclists?

**Details Of Ethics Concerns:**

No Ethics Concerns.

---

> ### Author Response · Authors · 2025-11-24
> **Response to Reviewer e7Me**
>
> We sincerely thank reviewer e7Me for the thorough and thoughtful assessment of our work. We appreciate the acknowledgement of our contributions as well as the constructive feedback on the strengths, possible limitations, and remaining questions in our approach. The comments have been extremely helpful in significantly improving our manuscript.
>
> Please note that we have uploaded our revised manuscript with changes highlighted in blue.
>
> ### Q-1 Clarification on novelty and differentiation from SN?
>
> **Novelty of TTSN:** To our knowledge, TTSN is the first test-time domain adaptation method that maintains stable and competitive accuracy without accessing source data or performing full model fine-tuning, thereby avoiding the severe degradation seen in OT.
>
> **TTSN vs SN:**
>
> - *Data Requirement:* Unlike SN, TTSN does not require access to the source domain dataset. Access to this data is often limited in practical applications due to the high cost associated with data labeling, which can discourage open-sourcing. TTSN can operate effectively using only unlabeled target domain data.
>
> - *Computational Efficiency:* SN necessitates modifying the source dataset (by scaling labels and point clouds) and then fully retraining the network on this modified data, which is both compute-intensive and time-consuming.TTSN avoids the bottleneck of retraining or finetuning the model to correct dimension bias. Instead, it addresses this bias during the test phase, making the process more cost and time-efficient.
>
> **Novelty of LLS and WSFT:** LLS and WSFT are introduced as lightweight, practical additions that offer resource-efficient solutions and broaden the overall framework beyond prior SN and OT approaches.
>
> Together, these methods form a comprehensive and efficient framework rather than just a simple extension of existing ideas.
>
> ### Q-2 & W-2 Reported results are only for the vehicle class, not for other classes, such as pedestrians and cyclists.
>
> Our choice of using a car class for our study is rooted in the fact that classes such as pedestrians and cyclists do not have much variance in size across regions and may not cause degradation in detection accuracies during adaptation, unlike the car class, where the huge variation in car size across regions is a driving factor behind detection quality degradation.
> We refer to Table S16 (as given below) from [1] and observe that there are smaller differences among datasets for the pedestrian class.
>
> |   | KITTI      | Argoverse | nuScenes  | Lyft      | Waymo     |
> |---|------------|-----------|-----------|-----------|-----------|
> | H | 1.76±0.11  | 1.84±0.15 | 1.78±0.18 | 1.76±0.18 | 1.75±0.20 |
> | W | 0.66±0.14  | 0.78±0.14 | 0.67±0.14 | 0.76±0.14 | 0.85±0.15 |
> | L | 0.84±0.23  | 0.78±0.14 | 0.73±0.19 | 0.78±0.17 | 0.90±0.19 |
>
>
> ### W-1 Effect of other deviations other than vehicle size
>
> We appreciate the reviewer for raising this important and valid point. Our current work primarily addresses the performance degradation caused by size differences across domains and regions, which we have identified as a major contributing factor to the decrease in performance. This is supported by the observation that applying our proposed methods to mitigate this dimension bias yields performance comparable to that of a detector specifically trained on the target domain (as shown by the 'target scores' in Tables 2 and 3). We certainly agree that other factors may exist and could be explored to achieve even greater performance improvements. We look forward to investigating these research avenues in our future work.

---

### Official Review · Reviewer_9eX6 · 2025-10-31

**Soundness:** 3
**Presentation:** 4
**Contribution:** 3
**Rating:** 6
**Confidence:** 4

**Summary:**

The research direction of this paper is domain adaptation in the 3D object detection field. To address the issue of detector performance degradation caused by significant differences in vehicle size between the source and target domains, three different resource scenarios for domain adaptation are proposed. TTSN requires only a small amount of unlabeled target domain data and average car size information to account for size variations during testing. LLS is suitable for cases with a small amount of labeled target domain data but difficulty in fine-tuning. WSFT uses a large amount of unlabeled target domain data along with their size information to calibrate the prediction box size using CSA loss.

**Strengths:**

1.Both TTSN and LLS methods use a small amount of unlabeled data from the target domain, reducing dependence on large amounts of data.

2.Divide into three scenarios according to the size of data volume and whether there are constraints on labels, respectively, and provide corresponding large-scale adaptation methods with high availability.

**Weaknesses:**

1.Experiments were conducted on datasets with significant differences in car size, but have you considered verifying the effectiveness of the method on smaller size difference datasets, such as from Waymo to Nuscenes?

2.For categories with smaller size differences, it is necessary to verify the effectiveness of this method.

**Questions:**

Is it feasible in reality to obtain statistical information about vehicles? Is the domain adaptation effective for unknown scenarios in practical applications? Where is the lightweight aspect reflected in the LLS method?

---

> ### Author Response · Authors · 2025-11-24
> **Response to Reviewer 9eX6**
>
> We would like to sincerely thank the reviewer 9eX6 for their careful and insightful assessment of our work. We appreciate the recognition of our contributions and the constructive feedback provided regarding the strengths, potential limitations, and open questions of our approach. The reviewer’s comments have been highly valuable in helping us further clarify the motivation, applicability, and practical considerations of our proposed methods. Below, we address each of the reviewer’s observations and questions in detail.
>
> Please note that we have updated our manuscript and highlighted the changes in blue.
>
> ### Q-1 Is it feasible in reality to obtain statistical information about vehicles?
>
> Using target car-size statistics is a practical substitute for 3D bounding box labels because these dimensions are easy to obtain from public vehicle data and align well with dataset-derived measurements. Such statistics can be sourced by examining publicly available data on commonly sold vehicle models (available at [1]) and averaging their reported dimensions (available at [2]). Published averages from large car markets in prior work closely match the dimensions computed directly from datasets like Waymo. This approach is both reliable and cost-effective, while maintaining a fair comparison with existing methods.
>
> ### Q-2 Is the domain adaptation effective for unknown scenarios in practical applications?
>
> Our methods specifically address the degradation in in-class performance caused by bounding box measurement errors that occur during the adaptation from the source to the target domain. Our experiments show that our approaches effectively mitigate these effects. Regarding the detection of object classes the model was not initially trained on (unknown objects), this challenge falls under the domain of continual learning, where a model incrementally learns new object classes within the target domain. This is an exciting and important direction for future 3D detection research that we are keen to explore.
>
> ### Q-3 Where is the lightweight aspect reflected in the LLS method?
>
> LLS employs scale factors (one for each of the three dimensions) to align the model predictions' dimensions with the target domain's ground truth during testing. To determine these factors, we train a small, lightweight regression network with only three parameters (the scale factors themselves) on a small labeled dataset. This approach is significantly more lightweight than few-shot fine-tuning, which requires training millions of parameters across the entire network. The advantage is that a user can train this small regression network efficiently on a CPU, eliminating the need for a GPU, which makes the process both cost-effective and time-saving.
>
> ### W-1 & W-2 Effectiveness of the method while adapting to datasets with smaller differences in statistics
>
> Thank you for raising this point. We have demonstrated the effectiveness of our method on the KITTI to nuScenes Domain Adaptation (DA) task. This task represents a scenario with smaller statistical differences between the source and target domains, a difference that is even smaller than in the Waymo-to-nuScenes task. The improvements shown in our KITTI to nuScenes adaptation results (see Table 3) directly address your concern.
>
>
> **References:**
>
> [1]  https://www.goodcarbadcar.net/2025-u-s-auto-sales-figures-by-model-all-vehicle-ranked-2
>
> [2]  https://www.carsized.com/en/cars/

---

### Official Review · Reviewer_9Yr3 · 2025-10-31

**Soundness:** 3
**Presentation:** 3
**Contribution:** 1
**Rating:** 2
**Confidence:** 4

**Summary:**

This paper tackles source-free domain adaptation for LiDAR 3D detection for autonomous driving: a test-time statistical normalization (TTSN), a tiny linear scaling layer (LLS) trained on few labeled samples, and a weakly supervised fine-tuning (WSFT) with a composite loss aligning size statistics while regularizing to source outputs.

**Strengths:**

1. The methods are intuitive and reasonable.

2. On authors' experiments, the proposed methods have proved effectiveness.

**Weaknesses:**

1. Novelty is limited; methods mainly recalibrate box dimensions and extend prior SN/OT/ST3D ideas.

2. Heavy reliance on car-size statistics.

3. Only one model is evaluated in the experiments. The proposed method may lack generalization to other 3D detection methods.

**Questions:**

Please see the weaknesses. My major concern is with the novelty of this paper. Please try to ditinguish your method with existing solutions.

---

> ### Author Response · Authors · 2025-11-24
> **Response to Reviewer 9Yr3**
>
> We would like to begin by expressing our sincere gratitude to Reviewer 9Yr3 for their thorough and thoughtful evaluation of our work. Their constructive feedback has been invaluable in helping us significantly improve and refine our paper. Below, we address their concerns point by point.
>
> Please note, we have uploaded a revised manuscript with the changes highlighted in blue.
>
> ### W-1 and Q-1 Novelty is limited; methods mainly recalibrate box dimensions and extend prior SN/OT/ST3D ideas.
>
> Thank you for this question. First, let us attempt to understand why recalibrating the box dimension is a very important problem.
>
> We agree that our main aim to fix the 3D bounding box dimensions and believe that this is a very important and crucial task for an autonomous system, having smaller or larger bounding boxes compared to the reality may cause severe issues as this is one of the foundational task and errors in this stage will propagate to later stages and even smaller errors can magnify issues, Resolving this issue brings target-domain performance close to that of full supervised training, highlighting the significance of the problem itself.
> In terms of novelty, TTSN is, to our knowledge, the first test-time approach that achieves stable and competitive Domain Adaptation (DA) performance without access to source data or full network fine-tuning, directly overcoming the major degradation issues present in OT, its only direct competitor.
>
> Next, we discuss what makes the proposed method distinct from existing SN/OT/ST3D ideas.
>
> **TTSN Novelty:** To the best of our knowledge, TTSN is the first test-time Domain Adaptation (DA) approach to achieve stable, competitive performance without requiring source data or full network fine-tuning, thereby solving the major degradation issues faced by its competitor, OT.
>
> **TTSN vs. OT Calibration:**
>
> - TTSN: Calculates calibration based on the difference between average model predictions on target data and actual target statistics.
>
> - OT: Calculates calibration simply from the difference between actual source and target statistics.
>
> **TTSN Advantage:** Accounts for how the source detector's predictions shift slightly toward target object sizes when applied to the target domain, a factor OT disregards.
>
> **Comprehensive Framework:** Further, we introduce LLS and WSFT to provide practical, resource-limited solutions, creating an efficient and comprehensive framework that extends beyond existing SN/OT work.
>
> **Comparison against ST3D:** ST3D requires access to the source data to pretrain the detector with random object scaling, whereas our methods do not rely on the source domain dataset for domain adaptation.
>
> Together, these methods form a comprehensive and efficient framework rather than a small extension of existing ideas.
>
> ### W-2 Heavy reliance on car-size statistics.
>
> We agree that our methods rely on target car-size statistics, and we chose this prior deliberately because it offers a simple, easily obtainable, and domain-relevant proxy for ground truth dimensions. Such statistics can be sourced by examining publicly available data on commonly sold vehicle models (available at [1]) and averaging their reported dimensions (available at [2]), which makes this approach practical in scenarios where annotated 3D bounding boxes are unavailable.
> Given both the accessibility of this information and the clear performance gains demonstrated in our results, we believe that using these statistics is a reasonable and cost-effective substitute for target-domain labels in point cloud datasets. We sincerely appreciate the reviewer for prompting us to clarify this design choice.
>
> ### W-3 Only one model is evaluated in the experiments. The proposed method may lack generalization to other 3D detection methods.
>
> To verify the generalizability, we conducted experiments using our methods with SECOND-iou and share our results below.
>
> **Scores on SECOND-iou model for Waymo to KITTI DA task:**
>
> | Methods | AP_BEV (Easy) | AP_BEV (Moderate) | AP_BEV (Hard) | AP_3D (Easy) | AP_3D (Moderate) | AP_3D (Hard) |
> |---------|:-------------:|:-----------------:|:-------------:|:------------:|:----------------:|:------------:|
> | DT      | 86.7858       | 66.5078           | 65.8248       | 54.4433,     | 40.7784          | 39.7578      |
> | OT      | 10.9822       | 11.7421           | 12.7272       | 0.0529       | 0.0529           | 0.0529       |
> | TTSN    | **89.1247**       | **68.5167**           | **68.1219**       | 70.8798      | 47.7474          | 46.8137      |
> | LLS     | 88.8150       | 68.2064           | 67.9509       | **73.2751**      | **52.2872**          | **48.5647**      |
>
> **References:**
>
> [1]  https://www.goodcarbadcar.net/2025-u-s-auto-sales-figures-by-model-all-vehicle-ranked-2
>
> [2]  https://www.carsized.com/en/cars/

---

> > ### Comment · Reviewer_9Yr3 · 2025-11-27
> >
> > Thanks for the rebuttal.
> >
> > The authors have addressed my concerns regarding the proposed method's generalization capability. The heavy reliance on car-size statistics has also been mitigated.
> >
> > Regarding the novelty, after checking related literature once again, I do agree with the authors: _"TTSN is the first test-time Domain Adaptation (DA) approach to achieve stable, competitive performance without requiring source data or full network fine-tuning, thereby solving the major degradation issues faced by its competitor, OT"._ However, I still feel that many prior works have explicitly considered similar ideas, but just adopted more complex methods to solve the problem. An example would be "Unsupervised Domain Adaptive 3D Detection With Multi-Level Consistency, ICCV 2021". The proposed method appears quite straightforward and resembles an engineering solution.
> >
> > --------
> >
> > In summary, I appreciate the authors' rebuttal and acknowledge the contribution of this work. I still have a bit concerns over the novelty. After careful consideration, I decided to revise my rating from 2 to 6.

---

> > > ### Author Response · Authors · 2025-11-27
> > > **Thank you Reviewer 9Yr3**
> > >
> > > We sincerely thank reviewer 9Yr3 for taking the time to evaluate our rebuttal. We are glad that our clarifications addressed your concerns regarding the generalization capability of our method and reliance on car-size statistics, and we appreciate your careful reassessment of the methodological novelty.
> > >
> > > Regarding the similarity to prior ideas in the literature, we agree that several existing works investigate related high-level concepts, often through more complex pipelines. Our goal with TTSN was to identify a simple yet effective mechanism that mitigates the major degradation issues observed in earlier test-time DA approaches, while eliminating the need for source data or extensive fine-tuning. Thank you for pointing out *Unsupervised Domain Adaptive 3D Detection With Multi-Level Consistency, ICCV 2021*. We will cite this work in our future revision.
> > >
> > > Thank you again for your constructive input and for updating your assessment. We genuinely appreciate your thoughtful consideration and feedback, which will help us further improve the clarity of our work.

---

### Official Review · Reviewer_VErs · 2025-11-02

**Soundness:** 2
**Presentation:** 2
**Contribution:** 1
**Rating:** 2
**Confidence:** 3

**Summary:**

This paper proposes three techniques to mitigate performance degradation from domain shift in LiDAR-based 3D object detection for autonomous driving. 1. TTSN (Test-Time Statistical Normalization): At inference, normalizes box dimensions using a correction derived from the gap between the target domain’s average car sizes (height/width/length) and the model’s predicted size statistics on target data. The correction is estimated from about 1K unlabeled target samples; no parameter updates are performed. 2. LLS (Lightweight Linear Scaling): Learns lightweight scaling coefficients (e.g., h/w/l) from a small set of labeled target samples to reduce dimensional mismatch. 3. WSFT (Weakly-Supervised Fine-Tuning): Fine-tunes without target labels using a Composite Statistical Alignment (CSA) loss combining a statistical alignment term on predicted sizes and a consistency term with the source model’s predictions, optimized with standard SGD. Models trained on KITTI are evaluated under domain shifts to nuScenes, Waymo, and Argoverse.

**Strengths:**

1. The problem formulation and deployment scenario are realistic. The paper explicitly considers source-free access, label-free or few-shot target supervision due to the cost of point-cloud annotation, and the practical burden/risks of full fine-tuning. The staged strategy (TTSN → LLS → WSFT) is motivated by these constraints.

2. The design is concise and staged. Focusing on box-dimension mismatch as the primary driver of degradation, the methods progress from inference-time calibration to lightweight scaling to weakly supervised fine-tuning, with adaptation strength and resource requirements (parameter updates, labels, target data) increasing accordingly. The prerequisites for each stage are clearly stated.

**Weaknesses:**

1. Contrary to the stated “consistent improvements,” Tables 2–3 do not consistently establish superiority. Beyond DT/OT, the effectiveness of the proposed methods should be established against contemporary TTA/SFDA baselines; such comparisons are not reported. While presented as source-free/training-free, the methods rely on a target-domain prior (car-size statistics). As a second-best option for fair comparison under equivalent resource assumptions, a direct comparison to FS (e.g., 10-shot) would help contextualize the gains. In the current tables, aside from modest gains on some KITTI→nuScenes metrics, the methods are often below FS on KITTI→Waymo and KITTI→Argoverse, so the claim of consistent effectiveness is not substantiated.

2. Labels and values are inconsistent across tables. Table 2 is labeled KITTI→Waymo but matches Table 3’s KITTI→nuScenes. Lines 405–408 also suggest the top panels of Table 3 are swapped—please correct captions/headers. Even after relabeling, numeric conflicts remain (e.g., TTSN AP_3D(Hard) 25.5 in Table 2 but 16.1 in Table 3, implying that Gap Closed and related statements (46.6%, 33.4%, 62.3%) should be recomputed. DT AP_BEV differs slightly (47.4/24.6/25.5 vs 47.3/24.6/25.4). Please clarify whether this is due to rounding, seed variation, or evaluation scripts.

3. Dataset splits are insufficiently specified, limiting reproducibility. Although Table 1 indicates custom train/validation partitions, the paper does not provide concrete procedures. For example, Waymo is described as 26k/6k “parsed” from the full set; nuScenes and Argoverse also depart from the official splits. Selection criteria (scene/city distribution, time/weather, sensor version, log-level sampling) and reproducible identifiers (frame/log ID lists, seeds) are not provided. The selection rule for the 1K calibration set used by TTSN (random vs. stratified, scene balance, disjointness from validation) is also unspecified. As a result, fair comparison and exact reproduction are difficult.

4. Ablations, analysis, and hyperparameter justifications are underdeveloped. Sec. 4.1 largely restates tables without explaining cross-shift behavior (e.g., TTSN strong on KITTI→nuScenes but behind LLS/WSFT on Waymo/Argoverse). The sole qualitative figure compares only to DT, omitting differences among TTSN/LLS/WSFT and failure/boundary cases. Key settings (WSFT’s CSA weight α, consistency IoU=0.3) lack rationale and sensitivity. Given the reliance on pseudo-label filtering, please at minimum report: (i) consistency on/off, (ii) results under different IoU thresholds (0.1/0.3/0.5), and (iii) results for different calibration-set sizes K (200/500/1k/2k). Without these, it is hard to assess whether the chosen settings are sound or if stronger configurations exist.

**Questions:**

1. You note that target car-size statistics (μh/μw/μl) can be obtained from public sources, yet in the experiments μ is computed from the dataset. Do you have results using truly public statistics, and can you share a deployment-ready procedure (sources, preprocessing, validation) and sensitivity to errors in μ?
2. Could you clarify why you did not use the official splits and provide the concrete splitting procedures for each dataset (train/val ratios, selection criteria, sampling rules, frame, seeds)?
3. Given that better alignment to target average car size should improve performance, how do you explain the substantial 3D drop of OT vs. DT in Table 2? To aid interpretation, would you share per-class and per-distance/size-bin metrics in addition to the aggregate scores?

---

> ### Author Response · Authors · 2025-11-24
> **Response #1 to Reviewer VErs**
>
> We would like to express our heartfelt gratitude to Reviewer VErs for their comprehensive and insightful evaluation of our work. Their detailed and constructive feedback has been instrumental in enhancing the quality of our paper. Below, we address each of their concerns in detail.
>
> Please note that we have uploaded a revised manuscript with edits highlighted in blue.
>
> ### Q-1: Difference between using target car-size statistics from public sources and the dataset stats & sensitivity to estimation error?
>
> Ans. Prior works, such as SN/OT, rely on target car-size statistics derived from the dataset itself. To ensure a fair comparison, we also compute these statistics directly from the dataset. However, [1] conducts an ablation study (Appendix S5.2) comparing the parameter ‘u’ computed from dataset-derived statistics versus publicly available data. Specifically, [1] reports average car dimensions (height, width, length) in the U.S. car market as [1.75, 1.93, 5.15], based on car sales data, which closely aligns with the statistics computed from the Waymo dataset, [1.80, 2.09, 4.97].
>
> **TTSN Results with real data (KITTI to Waymo)**
>
> |     Source     | AP_BEV (Easy) |   AP_BEV (Moderate)      |   AP_BEV (Hard)    | AP_3D (Easy) |   AP_3D (Moderate)      |   AP_3D (Hard)   |
> |:--------------:|:------:|:--------:|:----:|:-----:|:--------:|:----:|
> |       DT       |  41.0  |   35.6   | 36.8 |  10.4 |   10.6   | 10.4 |
> | Dataset        | 89.5   | 75.8     | 70.7 | 65.8  | 46.9     | 44.4 |
> | Car Sales data | 89.4   | 75.7     | 70.7 | 65.8  | 46.8     | 44.3 |
>
> We note that the performance remains largely unchanged regardless of the source of statistics.
>
> **TTSN Sensitivity to car stat estimation (KITTI to Waymo):**
>
> Over-estimation (w.r.t dataset-derived stats):
>
> | % Error       | AP_BEV (Easy) | AP_BEV (Moderate) | AP_BEV (Hard) | AP_3D (Easy) | AP_3D (Moderate) | AP_3D (Hard) |
> |---------------|:-------------:|:-----------------:|:-------------:|:------------:|:----------------:|:------------:|
> | Dataset stats | 89.5          | 75.8              | 70.7          | 65.8         | 46.9             | 44.4         |
> | 1% error      | 89.4322       | 75.6199           | 70.5767       | 59.9123      | 40.6414          | 39.7615      |
> | 3% error      | 88.6811       | 74.8342           | 69.8953       | 37.0049      | 27.2161          | 28.3458      |
> | 5% error      | 87.0976       | 68.7395           | 68.4371       | 20.7835      | 15.1626          | 16.6430      |
> | 8% error      | 79.9172       | 63.4431           | 62.5029       | 5.5431       | 4.5635           | 5.6049       |
> | 10% error     | 65.9487       | 55.1620           | 52.6801       | 1.5357       | 1.8525           | 2.2504       |
>
> Under-estimation (w.r.t dataset-derived stats):
>
> | % Error       | AP_BEV (Easy) | AP_BEV (Moderate) | AP_BEV (Hard) | AP_3D (Easy) | AP_3D (Moderate) | AP_3D (Hard) |
> |---------------|:-------------:|:-----------------:|:-------------:|:------------:|:----------------:|:------------:|
> | Dataset stats | 89.5          | 75.8              | 70.7          | 65.8         | 46.9             | 44.4         |
> | 1% error      | 89.4107       | 75.8162           | 70.7625       | 70.0048      | 50.6745          | 46.8199      |
> | 3% error      | 89.4793       | 75.9098           | 70.8392       | 72.8611      | 54.6972          | 52.7433      |
> | 5% error      | 89.2401       | 75.7002           | 70.6016       | 72.8548      | 55.1663          | 53.0722      |
> | 8% error      | 88.0019       | 70.1147           | 69.4751       | 66.0218      | 50.4400          | 46.1901      |
> | 10% error     | 84.4911       | 68.4747           | 67.1317       | 49.8002      | 38.9419          | 38.1767      |
>
> We observe that the performance drops linearly when the mean car size statistics are overestimated. Interestingly, a slight underestimation of these statistics leads to a performance improvement, but a large underestimation causes a performance drop. We attribute this behavior to the bias-variance tradeoff.
>
> Pseudo labels (predictions of the source model on the target domain data) consist of a consistent bias and random noise. Our mean size correction aims to eliminate the bias, but this process also increases the noise. A small underestimation subtly reduces the correction, which still removes most of the bias while injecting less noise, thus improving performance. Conversely, overestimation always overshoots the true value, worsening both the bias and the noise. A significant underestimation leaves too much bias uncorrected, which is why performance decreases in that scenario. This observed trend reflects the balance between effective bias removal and minimal noise amplification.
>
> We have discussed these results in Appendix A1 of the revised manuscript.

---

> ### Author Response · Authors · 2025-11-24
> **Response #2 to Reviewer VErs**
>
> ### Q-2  Clarification on data splits?
>
> For the KITTI dataset, we adopt dataset splits consistent with [1] (Sec. 3). For Argoverse and nuScenes, we use the training and validation splits after filtering for frames containing at least one car object, following the same procedure as [1] (details in Appendix S1.1). Our pre-filtered train/validation splits derived from the datasets match those reported in [1]: 13,122 / 5,015 for Argoverse and 11,040 / 3,026 for nuScenes.
>
> For the Waymo dataset, reproducing the subsampled dataset used in [1] was not feasible due to non-reproducibility issues in the public convert/waymo2kitti script provided at [2]. Specifically, the line
>
> `files = glob.glob(os.path.join(waymo_path, "training", "*.tfrecord"))`
>
> does not guarantee deterministic ordering of frame IDs, as it retrieves file paths in an arbitrary sequence. To address this, we instead used the validation split provided at [3], which contains 40,077 samples. We selected the validation split rather than the training split due to storage constraints (approximately 900 GB for training vs. 200 GB for validation). After filtering this set for frames containing car objects, we divided it into train/validation splits of 26,000 / 6,000 samples.
>
> We will release all filtered and unfiltered splits, along with their corresponding frame IDs, to ensure reproducibility. Furthermore, we resolved the aforementioned frame ID ordering issue during TFRecord extraction by enforcing a deterministic file order using the following modification:
>
> `files = sorted(glob.glob(os.path.join(waymo_path, "training", "*.tfrecord")))`
>
>
> ### Q-3. (a) Explain 3D drop in OT in Table 2. (b) Provide per-class and per-distance metrics for OT and TTSN?
>
> a) The authors [1] provide a reasonable interpretation for the score degradation (Sec. 5), stating that this might be due to over-correction of dimension bias: “this approach does not always improve but sometimes degrades the accuracy. This is because when we apply the source detector to the target domain, the predicted box sizes do slightly deviate from the source statistics to the target ones due to the difference in object sizes in the input signals (see Figure 4). Thus, simply adding (∆h, ∆w, ∆l) may over-correct the bias”.
>
> This actually served as motivation for the TTSN method, where our aim was to find a suitable scale for addition, which we achieved using pseudo-labels and a calibration vector.
>
> b) We only evaluate the Car class in our experiments, hypothesising that there isn’t much dimension variation in other classes, such as pedestrians/cyclists across regions. A proof for this argument is Table S16 in the Appendix of [1].
>
> Below, we show the scores of DT/OT/TTSN across the distance of the object from the autonomous car for the KITTI to nuScenes DA task.
>
> | Methods | AP_BEV (0-30m) | AP_BEV (30m-50m) | AP_BEV (50m-70m) | AP_3D (0-30m) | AP_3D (30m-50m) | AP_3D (50m-70m) |
> |---------|:--------------:|:----------------:|:----------------:|:-------------:|:---------------:|:---------------:|
> | DT      | 47.9           | 9.8              | 1.1              | 14.9          | 4.5             | 0.0             |
> | OT      | 56.2           | 10.8             | 1.5              | 13.9          | 9.1             | 1.0             |
> | TTSN    | 63.1           | 11.0             | 2.3              | 30.1          | 2.3             | 0.4             |
>
>
> **References:**
>
> [1] Yan Wang, Xiangyu Chen, Yurong You, Li Erran Li, Bharath Hariharan, Mark Campbell, Kilian Q Weinberger, and Wei-Lun Chao. Train in Germany, test in the USA: Making 3d object detectors generalize. In Proceedings of the IEEE/CVF conference on computer vision and pattern recognition, pp. 11713–11723, 2020b.
>
> [2] https://github.com/cxy1997/3D_adapt_auto_driving

---

> ### Author Response · Authors · 2025-11-24
> **Response #3 to Reviewer VErs**
>
> ### W-1 Lack of consistent superiority and insufficient comparison with contemporary baselines under equivalent resource assumptions.
>
> We sincerely appreciate the reviewer’s observation regarding the gains relative to DT/OT and the request for comparisons against contemporary TTA/SFDA methods. We would like to clarify the competitive landscape for each of our proposed components. For TTSN, the most directly comparable method is OT, which, to the best of our knowledge, is the only relevant TTA baseline that similarly relies on target-domain priors. Across all settings, TTSN consistently improves upon OT.
>
> For LLS, since the method uses a small amount of labeled target data and requires very limited computational cost (only three scale factors are optimized), we believe the most appropriate comparison is with FS, which also relies on limited labeled data but entails substantially higher computational requirements due to full-network finetuning. LLS outperforms FS on both KITTI→Waymo and KITTI→nuScenes, except in the moderate and hard cases of KITTI→Waymo, where performance remains very close. We hope the reviewer agrees that these small gaps should not detract from recognizing a substantially more efficient alternative.
>
> For WSFT, the method trains on pseudo-labels generated from a large set of unlabeled target samples using our proposed loss. The most suitable baseline here is pseudo-label–based self-training. In response to the reviewer’s request, we conducted new experiments and report AP_3D scores for Waymo→KITTI and nuScenes→KITTI, as these are the settings evaluated in [3].
>
> **AP_3D (Waymo→ KITTI)**
>
> | Method                      | AP_3D (Easy) | AP_3D (Moderate) | AP_3D (Hard) |
> |-----------------------------|:------------:|:----------------:|:------------:|
> | Pseudo-label self-training  | 21.1         | 19.3             | 18.3         |
> | WSFT (ours)                 | 18.5         |  26.5            | 31.6         |
>
>
> **AP_3D (nuScenes → KITTI)**
>
> | Method                      | AP_3D (Easy) | AP_3D (Moderate) | AP_3D (Hard) |
> |-----------------------------|:------------:|:----------------:|:------------:|
> | Pseudo-label self-training  |      22.2        | 11.6             | 11.9         |
> | WSFT (ours)                 | 23.2         | 23.8             | 26.4         |
>
> ### W-2 Issue with table labels and values
>
> This was a human error on our part, and the discrepancy in values is due to a rounding error. We have corrected the numeric conflict in TTSN AP_3D(Hard) score, updating it to 16.1. We have recomputed and verified the gap-closed percentage again. We are sincerely grateful to the reviewer for bringing this to our attention, and we have updated the manuscript with the corrected table labels and values.
>
> ### W-3 Clarification on data splits?
>
> We clarify this in the reply to question 2
>
> **References:**
>
> [3] Deepti Hegde and Vishal M. Patel. Attentive prototypes for source-free unsupervised domain adaptive 3d object detection. In Proceedings of the IEEE/CVF Winter Conference on Applications of Computer Vision (WACV), pp. 3066–3076, January 2024.

---

> ### Author Response · Authors · 2025-11-24
> **Response #4 to Reviewer VErs**
>
> ### W-4 (a) consistency on/off, (b) results under different IoU thresholds (0.1/0.3/0.5)
>
> We thank the reviewer for posing these important and valid questions regarding the choices for hyperparameters and for nudging us in the direction of exploring stronger settings.
>
> (a) For α, our choice reflects the degree to which we intend to penalize overfitting to the target-domain mean statistics. Because this term is intended solely as a regularizer, we found that setting α as high as 0.5 (giving it equal weight to the statistical alignment term) prevents the model from learning effectively. A smaller value provides the desired alignment toward the target distribution without forcing the predictions to collapse toward the mean.
>
> As asked by the reviewer, we conducted an ablation study to verify the usefulness of the consistency term by training with consistency off. The results on the validation set are presented below:
>
>
> | Epoch no.      | AP_BEV (Easy) | AP_BEV (Moderate) | AP_BEV (Hard) | AP_3D (Easy) | AP_3D (Moderate) | AP_3D (Hard) |
> |----------------|:-------------:|:-----------------:|:-------------:|:------------:|:----------------:|:------------:|
> | DT             | 41.0          | 35.6              | 36.8          | 10.4         | 10.6             | 10.4         |
> | Consistency on | 89.8          |  75.0             | 70.3          | 72.1         | 54.4             | 53.0         |
> | epoch 1        | 81.7913       | 68.8106           | 65.3187       | 40.9547      | 33.8467          | 35.4012      |
> | epoch 2        | 33.9005       | 33.7176           | 35.5122       | 6.4792       | 6.9373           | 13.6373      |
> | epoch 3        | 6.9481        | 8.0064            | 10.3212       | 0.3135       | 1.0101           | 1.0101       |
>
> We observe that without the consistency term, the model overfits to the target domain's mean statistics. This leads to a lower training loss but a decline in performance on the validation sets, ultimately collapsing after just three epochs as the model begins to predict only the mean dimension values. This demonstrates that the consistency term is crucial for preventing overfitting and thereby improving the source detector's performance.
>
> (b) On the IoU based filtering step in WSFT, we would like to clarify that this was a regrettable error on our side, the filtering step used to ensure that only high-quality pseudo labels are used for grounding was not based on IoU based pair filtering; instead, we were filtering the pseudo labels based on the logit score provided by the model before the greedy matching step. Our threshold for the logit score was 1, keeping in mind that higher thresholds yield cleaner pseudo labels but result in lower amount of pseudo labels available for the model to train on, whereas keeping a lower threshold on logit score (e.g <0.5) may result in low-quality pseudo-labels; both of these choices may result in a degradation in the model’s detection qualities.
>
> We conduct an ablation study on the KITTI to Waymo DA task, which is a direct parallel to the study on IoU filtering using the logit score threshold (most values fall in the range of [-5, 5]). We examine the results by keeping the logit score threshold for filtering at 0.3 and 2. The results are reported below:
>
> | threshold | AP_BEV (Easy) | AP_BEV (Moderate) | AP_BEV (Hard) | AP_3D (Easy) | AP_3D (Moderate) | AP_3D (Hard) |
> |-----------|:-------------:|:-----------------:|:-------------:|:------------:|:----------------:|:------------:|
> | 0.3       | 89.4685       | 74.3829           | 69.9520       | 70.2989      | 52.9459          | 51.7861      |
> | 1         | 89.8          | 75.0              | 70.3          | 72.1         | 54.4             | 53.0         |
> | 2         | 89.7488       | 75.3112           | 70.4943       | 71.6387      | 54.3091          | 52.3913      |
>
> We observe that in both directions, increasing and decreasing the threshold results in a degradation of the scores. Increasing the threshold yields a slight improvement in the bev metric, but threshold=1 remains the best overall choice for filtering.

---

> ### Author Response · Authors · 2025-11-24
> **Response #5 to Reviewer VErs**
>
> ### W-4 (c) results for different calibration-set sizes K (200/500/1k/2k).
>
> We conducted an ablation study on multiple sets of sizes [200,500,2k] to verify our choice for calibration-set size and to find if there exist any stronger settings  for TTSN on KITTI to Waymo DA task:
>
> | Calibration set size | AP_BEV (Easy) | AP_BEV (Moderate) | AP_BEV (Hard) | AP_3D (Easy) | AP_3D (Moderate) | AP_3D (Hard) |
> |----------------------|:-------------:|:-----------------:|:-------------:|:------------:|:----------------:|:------------:|
> | 200                  | 89.5282       | 75.7757           | 70.7250       | 64.6521      | 43.4128          | 43.7858      |
> | 500                  | 89.4925,      | 75.7777           | 70.7289       | 65.6740      | 46.7763          | 44.2776      |
> | 1k (curr)            | 89.5          | 75.8              | 70.7          | 65.8         | 46.9             | 44.4         |
> | 2k                   | 89.4925       | 75.7810           | 70.7320       | 65.8555      | 46.8938          | 44.3745      |
>
> We observe that the detection quality remains generally consistent across the {500, 1k, 2k} calibration set sizes. While the 200 set size shows a slight decrease in 3D scores, the 1k size achieves maximum performance. Choosing the 1k set size is therefore preferable, as it minimizes the requirement for unlabeled target domain samples without a significant drop in performance, especially when compared to the 2k size, where the performance gain does not justify the increased sample requirement.
>
>
> We sincerely appreciate the reviewer for their insightful and constructive feedback, which has been invaluable in significantly improving the quality of our manuscript. The specific comments highlighted critical errors that had previously eluded our attention, prompting us to conduct in-depth ablation studies. This helped us verify our hyperparameters and refine our analysis. We are highly grateful for the time and effort the reviewer dedicated to providing this guidance, and we look forward to any further discussion.

---

### Author Response · Authors · 2025-12-02
**Brief summary of rebuttal**

We want to express our sincere gratitude to the AC and the reviewers for their time and effort in evaluating our work. We appreciate all the constructive feedback, which has been instrumental in refining our manuscript, verifying our design choices through new ablation studies, and clarifying our contributions. We believe we have fully addressed the concerns raised and present a summary of our responses below. For a detailed discussion, please refer to our point-by-point response to each reviewer.

Please note that we have uploaded a revised manuscript containing new experiments conducted during the rebuttal process with edits highlighted in blue.

### [New Results]

**Generalization to New Architectures and Baselines**

Addressing Reviewer 9Yr3 and VErs comments, we have added new experiments to verify generalizability and clarify the competitive landscape:

- *SECOND-iou Architecture:* We added results using the SECOND-iou architecture on the Waymo to KITTI task, demonstrating that TTSN generalizes effectively beyond PointRCNN.

- *WSFT vs. Self-Training:* We added comparisons for WSFT against pseudo-label self-training (the most suitable baseline) on Waymo to KITTI and nuScenes to KITTI tasks

- We provided requested per-distance metrics for the KITTI to nuScenes task, demonstrating the effectiveness of TTSN in comparison to DT and OT across 0-30m, 30-50m, and 50-70m ranges.

**Ablation Studies**

Addressing Reviewer VErs comments, we have added detailed ablation studies to justify hyperparameters and analyze sensitivity:

- *Sensitivity to Car Statistics:* We added an analysis of TTSN's sensitivity to errors in car-size estimation.

- *Consistency Term:* Our new experiment, which excluded the consistency term from the loss function, demonstrated that removing this term leads to the model overfitting to the target mean statistics and training data.

- *Calibration Set Size:* We conducted a new experiment across different calibration set sizes and verified that a calibration set size of 1k is preferable, maximizing performance without the diminishing returns observed at 2k.

- *Logit Filtering:* We clarified that filtering is based on logit scores rather than IoU, and demonstrated, via an ablation study, that a threshold of 1 is optimal.

### [Clarity of Paper]

*Clarification on Novelty and Differentiation:* Addressing Reviewers 9Yr3 and e7Me, we explicitly clarified the distinctiveness of our contributions. We emphasized that TTSN is the first test-time DA method to achieve stability without access to source data or full fine-tuning, unlike SN, which requires source data, and OT, which suffers from degradation. We also highlighted that LLS and WSFT are designed as lightweight, resource-efficient additions to create a comprehensive framework.

*Data Splits:* We clarified our data split choices and provided reasoning for departing from the official splits. We have resolved non-reproducibility issues in prior public conversion scripts to ensure future reproducibility and will release all splits with frame IDs.

*Table Errors:* We corrected the human error regarding the numeric conflict in Table 2 and have updated the values and labels for Table 2 in our newly uploaded manuscript; the edits are highlighted in blue.

### [General comments]

We thank the reviewers for their helpful feedback and questions, which we believe have significantly improved our paper by strengthening our results through ablations and expanding its scope to additional architectures, such as SECOND-iou. We hope we were able to answer all the questions posed by the reviewers.

We would also like to highlight that Reviewer 9Yr3 was satisfied with our response and the additional experiments provided. Following our rebuttal, which clarified the novelty of our approach and demonstrated generalizability across architectures, *Reviewer 9Yr3 raised their score from 2 to 6*. We hope the AC will find these revisions compelling and evaluate the paper based on its merits.

---

### Meta-Review · Area_Chair_XUUU · 2025-12-25

**Summary:**

Three different techniques are presented to mitigate performance degradation from domain shifts in LiDAR-based 3D object detection for autonomous driving.  A lot more focus is presented on bounding box alignment.  This paper received mixed reviews.

**Reviewer Concerns:**

The core contributions (TTSN, LLS, WSFT) primarily address object size bias via post-hoc scaling, linear regression, or statistical alignment to mean car dimensions. These ideas closely follow prior work on statistical normalization and output transformation (e.g., Wang et al., 2020b), with limited methodological innovation beyond repackaging known calibration strategies under different resource constraints. The paper does not introduce new representations, learning principles, or theoretical insights into domain adaptation for 3D detection.

The work assumes that cross-domain degradation is dominated by vehicle size mismatch, while largely ignoring other critical factors such as point density differences, sensor noise characteristics, intensity distributions, occlusion patterns, environmental layout, and long-range sparsity. As a result, the proposed methods address only a single axis of domain shift, limiting their generality and scientific contribution.

Experiments focus almost exclusively on the Car class and rely heavily on datasets where size bias is already known to be a dominant factor. Performance gains largely vanish relative to the “Target” oracle, and improvements appear tied to correcting mean statistics rather than learning transferable structure. The paper does not convincingly demonstrate that the methods generalize to more complex shifts (e.g., class imbalance, rare objects, or non-rigid categories).

Especially in TTSN and LLS, adaptation occurs without modifying internal representations, relying instead on deterministic or linear corrections. Even WSFT optimizes a loss that enforces alignment to global statistics rather than learning discriminative, domain-invariant features. From a machine-learning perspective, the methods are shallow and closer to engineering heuristics than principled learning-based domain adaptation.

While framed as enabling “scalable, cross-regional deployment,” the paper does not analyze failure cases, safety implications, or downstream effects on planning and control. The reliance on external car-size statistics (e.g., registries or sales data) also weakens the claim of fully autonomous adaptation in real deployments.

**Reviewer Scores:**

While the authors have attempted to address some of the concerns raised by the reviewers, their response does not resolve the main issues outlined above. Although a few reviewers slightly increased their scores, their assessments were brief and expressed low confidence, which does not sufficiently mitigate the substantive concerns with the work.

---

### Decision · Program_Chairs · 2026-01-26

Reject